# DESTA: A Framework for Safe Reinforcement Learning with Markov Games of Intervention

## Abstract

Exploring in an unknown system can place an agent in dangerous situations, exposing to potentially catastrophic hazards. Many current approaches for tackling safe learning in reinforcement learning (RL) lead to a trade-off between safe exploration and fulfilling the task. Though these methods possibly incur fewer safety violations, they often also lead to reduced task performance. In this paper, we take the first step in introducing a generation of RL solvers that learn to minimise safety violations while maximising the task reward to the extent that can be tolerated by safe policies. Our approach uses a new two-player framework for safe RL called Distributive Exploration Safety Training Algorithm (DESTA). The core of DESTA is a novel design of game between two RL agents: Safety Agent that is delegated the task of minimising safety violations and Task Agent whose goal is to maximise the reward set by the environment task. Safety Agent can selectively take control of the system at any given point to prevent safety violations while Task Agent is free to execute its actions at all other states. This framework enables Safety Agent to learn to take actions that minimise future safety violations (during and after training) by performing safe actions at certain states while Task Agent performs actions that maximise the task performance everywhere else. We demonstrate DESTA's ability to tackle challenging tasks and compare against state-of-the-art RL methods in *Safety Gym Benchmarks* which simulate real-world physical systems and *OpenAI's Lunar Lander*.

## 1 Introduction

Reinforcement learning (RL) is a framework that enables autonomous agents to learn complex behaviours from interactions with the environment (Sutton & Barto, 2018). Inspired by concepts within neuroscience, RL has had notable successes in a number of practical domains such as robotics and video games (Deisenroth et al., 2011; Peng et al., 2017). During its training phase, an RL agent explores using a trial and error approach in order to determine the best actions. This process can lead to the agent selecting actions whose execution in some states may result in critical damage; for example, an aerial robot attempting to fly at high velocities can result in the helicopter crashing and subsequent permanent system failure. Consequently, a major challenge in RL is to produce methods that solve the task *and* ensure safety both during and after training.

A common approach to tackle the problem of safe exploration in RL is to use a constrained Markov decision process (c-MDP) formulation (Altman, 1998). In this framework, the agent seeks to maximise a single objective subject to various safety constraints. Although c-MDP can be solved if the model is known, extending this formalism to settings in which the model is unknown remains a significant challenge (Chow et al., 2019a). One of the main tools for tackling the c-MDP problem setting in RL is the Lagrangian approach for solving a constrained problem, that is, solving $\max_\theta \min_\lambda f(\theta) - \lambda g(\theta)$ by gradient descent in $\lambda$ and ascent in $\theta$ instead of solving $\max_\theta f(\theta)$ subject to $g(\theta) \leq 0$. This approach introduces a *trade-off* between *competing* objectives: maximising the task reward and minimising safety violation. This trade-off which is calibrated by $\lambda$, undermines the ability of the agent to optimise both criteria simultaneously (Srinivasan et al., 2020; Kahn et al., 2017). Consequently, the resulting policies do not offer guarantees of safety during training Platt & Barr (1987); Song & Leland (1998) and can lead to safety violations both during training and at convergence (Chow et al., 2018; Stooke et al., 2020).

In contrast, to ensure their safety, animals exhibit a vast number of *safety reflexes*: reactive intervention systems designed to override and assume control in dangerous situations to prevent injury. One such example is the *diving reflex*, a sequence of physiological responses to the threat of oxygen deprivation (asphyxiation) that overrides the body's basic behavioural and regulatory (homeostatic) systems. This reflex is fundamental that it has been found in all air-breathing vertebrates (Butler & Jones, 1997).

Inspired by such naturally occurring systems, in this paper, we tackle the challenge of learning safely in RL with a new two-agent framework for safe exploration and learning, DESTA. The framework entails an interdependent interaction between an RL agent, Task Agent whose goal is to maximise the set of environment rewards and an additional RL agent, Safety Agent whose goal is to ensure the safety of the system. The Safety Agent has the authority to override the Task Agent and *assume* control of the system and, using a *deterministic policy* to avoid unsafe states while Task Agent can perform its actions everywhere else to maximise rewards. As such, the goals of completing the task set by the environment and ensuring safety are decoupled and each is delegated to an individual agent.

Transforming these components into a workable framework requires a formalism known as *Markov games* (MGs). MGs augment MDPs to tackling strategic interactions between agents (Littman, 1994). To bridge the gap between safety solutions offered in nature and machine learning, we develop a new type of MG namely a *nonzero-sum MG of interventions*. In this novel type of MG, two agents with distinct goals *cooperate* and exchange control of the system to minimise safety violations meanwhile maximising the task objective. Our framework, which consists of a strategic interaction between two independent RL agents confers several key advantages:

- **Decoupled Objectives & Safety Planning:** The tasks of maximising the environment reward and minimising safety violations are *fully decoupled*. This means Safety Agent pursues its safety objectives without trading-off safety for environment rewards and, performs *safe planning* to minimise safety violations at future system states. Moreover, since the agents are trained in a shared system, each agent learns a policy which acts in response to the behaviour of the agent. In particular, Safety Agent anticipates Task Agent's future actions (after and *during training*) and acts to reduce safety violations (see Experiment 1).

- **Selective (deterministic) interventions:** Safety Agent acts *only* at states in which its action ensures the safety criteria. At states in which the safety criterion is irrelevant only actions that are relevant to the task are played. Moreover Safety Agent uses a *deterministic policy* which eliminates inadvertent actions that may lead to safety violations (see Lunar Lander experiment).

- **Plug & play:** DESTA is highly flexible and accommodates various notions of safety within Safety Agent's objective. Unlike various approaches in which safety contingencies are manually embedded into the policy e.g. Schulman et al. (2015), DESTA accommodates any RL policy. This also enables safe policies to be (re)used and applied only at states where necessary.

## 2 RELATED WORK

Recent works in safe RL include assumptions from knowing the set of safe states and access to a safe policy (Koller et al., 2018; Berkenkamp et al., 2017), knowing the environment model (Fisac et al., 2019; Dean et al., 2019), having access to the cost function (Chow et al., 2019b), having a continuous safety cost function (Cowen-Rivers et al., 2020) and using reversibility as a criterion for safety (Eysenbach et al., 2018). Constrained Policy Optimization (CPO) (Achiam et al., 2017) extends trust region policy optimisation (TRPO) (Schulman et al., 2015) with the aim of ensuring that a feasible policy stays within the constraints in expectation. Similarly in the context of multi-agent RL (Yang & Wang, 2020), Gu et al. (2021) develops MACPO by extending multi-agent TRPO (Kuba et al., 2021) with safe constraints. However, the convergence of these methods is still challenging; the learning dynamics tend to oscillate Stooke et al. (2020), and the methodology does not readily accommodate general RL algorithms (Chow et al., 2018). In Tessler et al. (2018), a reward-shaping approach is used to guide the learned policy to satisfy the constraints, however their approach provides no guarantees during the learning process. In Dalal et al. (2018), a safety layer is introduced that acts on top of an RL agent's possibly unsafe actions to prevent safety violations though their framework does not deal with negative long-term consequences of an action. In Bharadhwaj et al. (2021), a similar framework to CPO is used with a sparse binary safety signal where the Q function is overestimated to provide tuneable safety. Recently, Stooke et al. (2020) investigated the oscillation issue from a dynamical system point of view and introduced a treatment by applying a PID controller on the dual variable.

To combat the limitations of the c-MDP formulation, several methods transform the original constraint to a more conservative one to ease the problem. For example El Chamie et al. (2016) replace the constraint cost with a conservative step-wise surrogate constraint. A significant drawback of these approaches is their conservativeness undermines task performance (the extent of the sub-optimality has yet to be characterised). Other approaches manually embed engineered safety-responses that are executed near safety-violating regions (Eysenbach et al., 2018; Turchetta et al., 2020). For example, in Turchetta et al. (2020), a safe teacher-student framework in which the teacher's objective is the value of the student's final policy, the agent is endowed with a pre-specified library of reset controls that it activates close to danger. These approaches can require time-consuming human input contrary to the goal of autonomous learning. Moreover, these approaches do not allow the safety response to anticipate future the behaviour of the RL agent and do not perform safety planning.

## 3    PRELIMINARIES

**Reinforcement Learning (RL).** In RL, an agent sequentially selects actions to maximise its expected returns. The underlying problem is typically formalised as an MDP $\langle \mathcal{S}, \mathcal{A}, P, R, \gamma \rangle$ where $\mathcal{S} \subset \mathbb{R}^p$ is the set of states, $\mathcal{A} \subset \mathbb{R}^k$ is the set of actions, $P : \mathcal{S} \times \mathcal{A} \times \mathcal{S} \to [0, 1]$ is a transition probability function describing the system's dynamics, $R : \mathcal{S} \times \mathcal{A} \to \mathbb{R}$ is the reward function measuring the agent's performance and the factor $\gamma \in [0, 1)$ specifies the degree to which the agent's rewards are discounted over time (Sutton & Barto, 2018). At time $t \in 0, 1, \ldots$, the system is in state $s_t \in \mathcal{S}$ and the agent must choose an action $a_t \in \mathcal{A}$ which transitions the system to a new state $s_{t+1} \sim P(\cdot|s_t, a_t)$ and produces a reward $R(s_t, a_t)$. A policy $\pi : \mathcal{S} \times \mathcal{A} \to [0, 1]$ is a probability distribution over state-action pairs where $\pi(a|s)$ represents the probability of selecting action $a \in \mathcal{A}$ in state $s \in \mathcal{S}$. The goal of an RL agent is to find a policy $\hat{\pi} \in \Pi$ that maximises its expected returns given by the value function: $v^\pi(s) = \mathbb{E}[\sum_{t=0}^\infty \gamma^t R(s_t, a_t)|a_t \sim \pi(\cdot|s_t)]$ where $\Pi$ is the agent's policy set.

**Safety in RL.** A key concern for RL in control and robotics settings is the idea of safety. This is handled in two main ways (García et al., 2015): using prior knowledge of safe states to constrain the policy during learning or modifying the objective to incorporate appropriate penalties or safety constraints. The constrained MDP (c-MDP) framework (Altman, 1999) is a central formalism for tackling safety within RL. This involves maximising reward while maintaining costs within certain bounds which restricts the set of allowable policies for the MDP. Formally, a c-MDP consists of an MDP $\langle \mathcal{S}, \mathcal{A}, P, R, \gamma \rangle$ and $\mathcal{C} = \{(L_i : \mathcal{S} \times \mathcal{A} \to \mathbb{R}, d_i \in \mathbb{R})|i = 1, 2 \ldots n\}$, which is a set of safety constraint functions $\boldsymbol{L} := (L_1, \ldots L_n)$ that the agent must satisfy and $\{d_i\}$ which describe the extent to which the constraints are allowed to be not satisfied. Given a set of allowed policies $\Pi_C := \{\pi \in \Pi : v_{L_i}^\pi \leq d_i \; \forall i = 1, \ldots, n\}$ where $v_{L_i}^\pi(s) = \mathbb{E}[\sum_{t=0}^\infty \gamma^t L_i(s_t, a_t)|s_0 = s]$, the c-MDP objective is to find a policy $\pi^\star$ such that $\pi^\star \in \arg\max_{\pi \in \Pi_C} v^\pi(s)$, for all $s \in \mathcal{S}$. The accumulative safety costs can be represented using hard constraints, this captures for example avoiding subregions $\mathcal{U} \subset \mathcal{S}$. When $L_i$ is an indicator function i.e. takes values $\{0, 1\}$, it is easy to see that each $v_{L_i}$ represents the accumulated probability of safety violations since $v_{L_i}^\pi(s) = \mathbb{E}[\sum_{t=0}^\infty \gamma^t 1(s_t)|s_0 = s] = \mathbb{P}(violation)$.

*Safe exploration in RL* seeks to address the challenge of learning an optimal policy for a task while minimising the occurrence of safety violations (or catastrophic failures) during training (Hans et al., 2008). Since in RL, the model dynamics and reward function are a priori unknown, the aim is to keep the frequency of failure in each training episode as small as possible.

**Markov games.** Our framework involves a system of two agents each with their individual objectives. Settings of this kind are formalised by MGs, a framework for studying self-interested agents that simultaneously act over time (Littman, 1994). In the standard MG setup, the actions of *both* agents influence both each agent's rewards and the system dynamics. Therefore, each agent $i \in \{1, 2\}$ has its own reward function $R_i : \mathcal{S} \times (\times_{i=1}^2 \mathcal{A}_i) \to \mathbb{R}$ and action set $\mathcal{A}_i$ and its goal is to maximise its *own* expected returns. The system dynamics, now influenced by both agents, are described by a transition probability $P : \mathcal{S} \times (\times_{i=1}^2 \mathcal{A}_i) \times \mathcal{S} \to [0, 1]$. Unlike classical MGs, in our MG, Safety Agent does not intervene at each state but is allowed to assume control of the system at certain states which it decides using a form of control known as *impulse control* (Mguni, 2018).

# 4 OUR FRAMEWORK

We now describe our framework which consists of two core components: firstly an MG between two agents, Task Agent and a second agent, the Safety Agent and, an impulse control component which is used by the Safety Agent. The impulse control component allows the Safety Agent to be selective about the set of states that it assumes control (and in doing so influence the transition dynamics and reward) so that actions geared towards safety concerns are performed only at relevant states. This leaves Task Agent to maximise the environment reward everywhere else. Unlike the c-MDP formulation, the goal of minimising safety violations and maximising the task reward are now delegated to two individual agents that now have distinct objectives.

## 4.1 A MARKOV GAME OF INTERVENTIONS ON ONE SIDE

We introduce our new MG of interventions on one side. Formally, our MG is defined by a tuple $\mathcal{G} = \langle \mathcal{N}, \mathcal{S}, \mathcal{A}, \mathcal{A}^{2,\mathrm{safe}}, P, R_1, R_2, \gamma \rangle$ where the new elements are the set of agents $\mathcal{N} = \{1, 2\}$, $\mathcal{A}^{2,\mathrm{safe}} \subseteq \mathcal{A}$ which is the action set for Safety Agent and the functions $R_i : \mathcal{S} \times \mathcal{A} \times \mathcal{A}^{2,\mathrm{safe}} \to \mathbb{R}$ the one-step reward for agent $i \in \{1, 2\}$. The transition probability $P : \mathcal{S} \times \mathcal{A} \times \mathcal{A}^{2,\mathrm{safe}} \times \mathcal{S} \to [0, 1]$ takes the state and action of both agents as inputs. Task Agent and Safety Agent use the Markov policies $\pi : \mathcal{S} \times \mathcal{A} \to [0, 1]$ and $\pi^{2,\mathrm{safe}} : \mathcal{S} \times \mathcal{A}^{2,\mathrm{safe}} \to [0, 1]$ respectively each of which is contained in the sets $\Pi$ and $\Pi^{2,\mathrm{safe}} \subset \Pi$ which are a stochastic policy set and a *deterministic policy* subset respectively. Therefore, whenever Safety Agent assumes control, random exploratory actions are switched off. Lastly, Safety Agent also has a (categorical) policy $\mathfrak{g}_2 : \mathcal{S} \to \{0, 1\}$ which it uses to determine whether or not it should intervene.

Denote by $\{\tau_k\}_{k \geq 0}$ the *intervention times* or points at which the Safety Agent decides to take an action so for example if the Safety Agent chooses to take an actions at state $s_6$ and again at state $s_8$, then $\tau_1 = 6$ and $\tau_2 = 8$ (we will shortly describe these in more detail). At any instance the transition dynamics are affected by only the Safety Agent whenever it decides to act (Task Agent influences the dynamics at all other times). With this in mind, define the function $\boldsymbol{P} : \mathcal{S} \times \mathcal{A} \times \mathcal{A}^{2,\mathrm{safe}} \times \mathcal{S} \to [0, 1]$ by $\boldsymbol{P}(s_{t+1}, a_t, \sum_{0 \leq k \leq t} a_t^{2,\mathrm{safe}} \delta_t^{\tau_k}, s_t)$ where the function $\delta_b^a$ is the Kronecker-delta function (so $\sum_{0 \leq k \leq t} a_t^{2,\mathrm{safe}} \delta_t^{\tau_k}$ is $a_{\tau_k}^{2,\mathrm{safe}}$ whenever $t = \tau_1, \tau_2, \ldots$ and null otherwise). The transition dynamics are determined by the probability transition function given by

$$\boldsymbol{P}(s', a, a^{2,\mathrm{safe}}, s) = P(s', a, s) \left( 1 - \mathbf{1}_{\mathcal{A}^{2,\mathrm{safe}}}(a^{2,\mathrm{safe}}) \right) + P(s', a^{2,\mathrm{safe}}, s) \mathbf{1}_{\mathcal{A}^{2,\mathrm{safe}}}(a^{2,\mathrm{safe}}). \quad (1)$$

Note that if Safety Agent plays a fixed policy then the MG reduces to an MDP.

## 4.2 THE TASK AGENT OBJECTIVE

The goal of Task Agent is to maximise its expected cumulative reward set by the environment (note that this does not include safety which is delegated to Safety Agent). To construct the objective for Task Agent, we begin by defining the function $R_1 : \mathcal{S} \times \mathcal{A} \times \mathcal{A}^{2,\mathrm{safe}} \to \mathbb{R}$ by $R_1(s_t, a_t, a_t^{2,\mathrm{safe}}) = R(s_t, a_t)(1 - \mathbf{1}_{\mathcal{A}^{2,\mathrm{safe}}}(a_t^{2,\mathrm{safe}})) + R(s, a_t^{2,\mathrm{safe}}) \mathbf{1}_{\mathcal{A}^{2,\mathrm{safe}}}(a_t^{2,\mathrm{safe}})$ where $\mathbf{1}_{\mathcal{Y}}(y)$ is the indicator function which is 1 whenever $y \in \mathcal{Y}$ and 0 otherwise. The objective that Task Agent seeks to maximise is:

$$v_1^{\pi, (\pi^{2,\mathrm{safe}}, \mathfrak{g}_2)}(s) = \mathbb{E} \left[ \sum_{t \geq 0} \sum_{0 \leq k} \gamma^t R_1 \left( s_t, a_t, a_{\tau_k}^{2,\mathrm{safe}} \right) \delta_t^{\tau_k} \Big| s_0 \equiv s \right], \quad (2)$$

where $a_t \sim \pi(\cdot|s_t)$ is Task Agent's action and $a_t^{2,\mathrm{safe}} \sim \pi^{2,\mathrm{safe}}(\cdot|s_t)$ is an action chosen by the Safety Agent. Therefore, the reward received by Task Agent is $R(s_t, a_t^{2,\mathrm{safe}})$ when $t = \tau_k$, $k = 0, 1, \ldots$ i.e. whenever the Safety Agent decides to take an action and $R(s_t, a_t)$ otherwise.

## 4.3 THE SAFETY AGENT OBJECTIVE

The goal of the Safety Agent is to minimise safety violations both during and after training. Unlike Task Agent whose actions incur no cost, for the Safety Agent, each intervention incurs a cost applied

by a cost function $c : \mathcal{A}^{2,\text{safe}} \to \mathbb{R}_{>0}$. The cost ensures any Safety Agent interventions are warranted by an increase in safety. The objective that Safety Agent seeks to maximise is:

$$v_2^{\pi,(\pi^{2,\text{safe}},\mathfrak{g}_2)}(s) = \mathbb{E}\left[\sum_{t \geq 0}\sum_{0 \leq k}\gamma^t\left(-\bar{\boldsymbol{L}}(s_t,a_t,a_{\tau_k}^{2,\text{safe}}) - c(a_{\tau_k}^{2,\text{safe}})\delta_t^{\tau_k}\right)\right]. \tag{3}$$

where $\bar{\boldsymbol{L}} : \mathcal{S} \times \mathcal{A} \times \mathcal{A}^{2,\text{safe}} \to \mathbb{R}$ is defined by $\bar{\boldsymbol{L}}(s_t,a_t,a_t^{2,\text{safe}}) = \boldsymbol{L}(s_t,a_t)(1 - \mathbf{1}_{\mathcal{A}^{2,\text{safe}}}(a_t^{2,\text{safe}})) + \boldsymbol{L}(s,a_t^{2,\text{safe}})\mathbf{1}_{\mathcal{A}^{2,\text{safe}}}(a_t^{2,\text{safe}})$ and $\boldsymbol{L} := (L_1, \ldots L_n)$ is a set of constraint functions that indicate how much a given constraint has been violated and is provided by the environment. Therefore to maximise its objective, the Safety Agent must determine the sequence of points $\{\tau_k\}$ at which the benefit of performing a precise action overcomes the cost of doing so. We specialise to the case $c(a^{2,\text{safe}}) = \kappa \cdot a^{2,\text{safe}}$ where $\kappa$ is a positive constant. Each function $L_i$ can represent a (possibly binary) signal that indicates a visitation to an unsafe state.

We now mention some key aspects of the framework. In particular, now the task of ensuring safety is delegated to Safety agent whose sole objective is minimise safety violations throughout the course of the problem. Secondly, the presence of the (strictly negative) intervention cost induces a selection process for Safety agent whereby it seeks only to intervene at the set of states for which the reduction in cumulative safety violations is sufficiently high to merit an intervention. This means that at all other states, task agent can freely act and in doing so learn to play actions that deliver task rewards whenever there is no potential for safety violations. Lastly, since the agents learn how to respond to one another, the strategic interaction between the two agents leads to policies in which task agent anticipates the actions of the task agent and vice-versa.

### 4.4 THE SAFETY AGENT IMPULSE CONTROL MECHANISM

The problem for the Safety Agent is to determine at which states it should assume control and what actions it should perform. We now describe how at a given state Safety Agent decides to intervene and overwrite Task Agent or not and the magnitudes of such interventions . In our setup at each state the Safety Agent first makes a *binary decision* to decide to *assume control*. Therefore, the *intervention times* $\{\tau_k\}$ *are **rules** that depend on the state* that are given by $\tau_k = \inf\{t > \tau_{k-1}|s_t \in \mathcal{T}, \mathfrak{g}_2(s_t) = 1\}$ where $\mathcal{T}$ is the or sequence of states induced by the joint actions of Task Agent and the Safety Agent and the probability kernel $P$. Therefore, by learning an optimal $\mathfrak{g}_2$, the Safety Agent learns the useful states to perform an intervention. As we later explain, these intervention times are determined by a condition on the state which is easy to evaluate (see Prop. 1).

Unlike (Eysenbach et al., 2018; Turchetta et al., 2020), our approach enables learning a safe intervention policy during training without the need to preprogram manually engineered safety responses and minimises safety violations during training unlike (Tessler et al., 2018; Dalal et al., 2018). Unlike (Fisac et al., 2019; Dean et al., 2019; Chow et al., 2019b; Fisac et al., 2019), which require access to information which is not available without a priori knowledge of the environment, our framework does not require a priori knowledge of the model of the environment or the unsafe states.

Learning to solve an MG involves finding a stable point in the independent agents' policies. In our MG, this means the Safety Agent learns to assume control at a subset of states and minimise safety violations given Task Agent's policy and, Task Agent learns to execute actions that maximise the task objective at all other states (given the actions of Safety Agent).

## 5 THE LEARNING METHOD

A key aspect of our framework is the presence of two RL agents that each adapt their play according to each other's behaviour. This produces two concurrent learning processes each designed to fulfill distinct objectives. At a stable point of the learning processes the Safety Agent minimises safety violations while Task Agent maximises the environment reward. Additionally, Safety Agent learns the set of states in which to perform an action to maintain safety at the current or future states. Therefore, central to this process is the decision of where Safety Agent should intervene. The intervention times are characterised by an 'obstacle condition' which can be evaluated online.

We now derive the condition that characterises Safety Agent's intervention times. We begin by first defining a key object in preparation for this characterisation:

---

**Algorithm 1:** **D**istributive **E**xploration **S**afety **T**raining **A**lgorithm (DESTA)

---

1: Initialise replay buffers $\mathcal{D} = \{\emptyset\}$, $\mathcal{D}^{2,\text{safe}} = \{\emptyset\}$, $\mathcal{D}^{\text{int}} = \{\emptyset\}$,
2: **for** $N_{episodes}$ **do**
3:     State $s_0$
4:     **for** $t = 0, 1, \ldots$ **do**
5:         Sample a task action $a_t \sim \pi(\cdot|s_t)$, a safe action $a_t^{2,\text{safe}} \sim \pi^2(\cdot|s_t)$, and an intervention
            action $a_t^{\text{int}} \sim \mathfrak{g}_2(\cdot|s_t) \, (\in \{0,1\})$
6:         **if** $a_t^{\text{int}} = 0$ **then**
7:             Apply task action $a_t$ so $s_{t+1} \sim P(\cdot|a_t, s_t)$. Set $a = a_t$
8:         **else if** $a_t^{\text{int}} = 1$ **then**
9:             Apply safe action $a_t^{2,\text{safe}}$ so $s_{t+1} \sim P(\cdot|a_t^{2,\text{safe}}, s_t)$. Set $a = a_t^{2,\text{safe}}$
10:         **end if**
11:         Receive reward $R(s_t, a)$ and cost $\boldsymbol{L}(s_t, a)$
12:         Set $R_1 = R(s_t, a)$, $R_2 = -\boldsymbol{L}(s_t, a) - c(a)$, $R^{\text{int}} = -\boldsymbol{L}(s_t, a) - c(a)a_t^{\text{int}}$
13:         Add the sample $(s_t, a, s_{t+1}, R_1)$ to $\mathcal{D}$, the sample $(s_t, a, s_{t+1}, R_2)$ to $\mathcal{D}^{2,\text{safe}}$, the sample
            $(s_t, a_t^{\text{int}}, s_{t+1}, R^{\text{int}})$ to $\mathcal{D}^{\text{int}}$
14:     **end for**
15:     **// Learn the individual policies**
16:     Update the policy $\pi$ using $\mathcal{D}$, the policy $\pi^{2,\text{safe}}$ using $\mathcal{D}^{2,\text{safe}}$, the policy $\mathfrak{g}_2$ using $\mathcal{D}^{\text{int}}$
17: **end for**

---

We first introduce the following object which is required for constructing the Bellman operator:

**Definition 1** *Let $\pi \in \Pi$ and $\pi^{2,\text{safe}} \in \Pi^{2,\text{safe}}$ be a Task Agent and a Safety Agent policy respectively, then for any $s_{\tau_k} \in \mathcal{S}$ and for any $\tau_k \in \tau_1, \tau_2, \ldots$, define by $\mathcal{Q}_2^{\pi,\pi^{2,\text{safe}}}(s_{\tau_k}, \pi_{\tau_k}^{2,\text{safe}}) :=$*
*$-\boldsymbol{L}(s_{\tau_k}, \pi_{\tau_k}^{2,\text{safe}}) - c(\pi_{\tau_k}^{2,\text{safe}}) + \gamma \int_{\mathcal{S}} ds P(s; \pi_{\tau_k}^{2,\text{safe}}, s_{\tau_k}) v_2^{\pi,\pi^{2,\text{safe}}}(s)$. We define the intervention*
*operator $\mathcal{M}^{\pi,\pi^{2,\text{safe}}}$ by the following: $\mathcal{M}^{\pi,\pi^{2,\text{safe}}} v_2^{\pi,\pi^{2,\text{safe}}}(s_{\tau_k}) := \max_{a' \in \mathcal{A}^{2,\text{safe}}} \mathcal{Q}_2^{\pi,\pi^{2,\text{safe}}}(s_{\tau_k}, a')$.*

The quantity $\mathcal{M}v_2$ measures the expected future stream of rewards for Safety Agent after an immediate (optimal) intervention minus the cost of intervening. The following result characterises the Safety Agent policy $\mathfrak{g}_2$ and the times that Safety Agent must perform an intervention.

**Proposition 1** *For any $s \in \mathcal{S}$, the Safety Agent intervention policy $\mathfrak{g}_2$ is given by the following expression $\mathfrak{g}_2(s) = H\left(\mathbb{E}_{a\sim\pi}\left[\mathcal{M}^{\pi^2} v_2^{\pi,\pi^{2,\text{safe}}}(s) - Q_2^{\pi,\pi^{2,\text{safe}}}(s,a)\right]\right)(s)$ where, $Q_2^{\pi,\pi^{2,\text{safe}}}(s,a) :=$*
*$-\boldsymbol{L}(s,a) + \gamma \int_{\mathcal{S}} ds' P(s';a,s) v_2^{\pi,\pi^{2,\text{safe}}}(s')$ and $H$ is the Heaviside function, moreover Safety Agent's intervention times are given by $\tau_k = \inf\left\{\tau > \tau_{k-1} | \mathcal{M}^{\pi^{2,\text{safe}}} v_2^{\pi,\pi^{2,\text{safe}}} = v_2^{\pi,\pi^{2,\text{safe}}}\right\}$.*

Hence, Prop. 1 also characterises the (categorical) distribution $\mathfrak{g}_2$. Moreover, the times $\{\tau_k\}$ can be determined by evaluating if $\mathcal{M}v_2 = v_2$ holds. The result yields a key aspect of our algorithm for executing Safety agent's activations.

While implementing the intervention policy appears to be straightforward by comparing value functions, it requires the optimal value functions in question. Furthermore, learning the intervention policy and these value functions simultaneously resulted in an unstable procedure. As a solution we propose to learn $\mathfrak{g}_2$ using an off-the-shelf policy gradient algorithm (such as TRPO, PPO or SAC). This policy is categorical with values $\{0,1\}$ and has a reward $R^{\text{int}}$ equal to $-\boldsymbol{L}(s_t,a) + c(a)$ if the safety agent intervenes with an action $a = a_t^{2,\text{safe}}$, and $-\boldsymbol{L}(s_t,a)$ if the safety agent does not intervene, i.e., the task agent applies an action $a = a_t$.

Implementing both the safety policy, which maximise the safety associated reward, and the task policy, which maximises the task reward, can be done using any off-the-shelf policy gradient methods without modifications. To summarise, we will learn three policies: the first one learning the actions of the task agent, the second learning the action of the safety agent, if the safety agent intervenes, and the final learning to intervene.

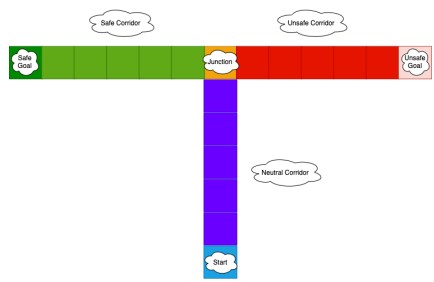

Figure 1: Diagram of the T-Junction environment

We take an off-policy approach and every policy is an instance of SAC with appropriate action spaces and rewards. Since we learn off-policy every agent can collect the same triplets $s_t$, $a$, $s_{t+1}$, where $a$ is the current action (i.e., either the task action $a_t$ or the safety action $a_t^{2,\text{safe}}$). The rewards that the policies will receive will differ though. The task policy will receive $R_1 = R(s_t, a)$, the safety policy will get $R_2 = -\boldsymbol{L}(s_t, a) - c(a_t)$, and the intervention policy rewards are $R^{\text{int}} = -\boldsymbol{L}(s_t, a) - c(a)a^{\text{int}}$, where $a^{\text{int}} \sim \mathfrak{g}_2(\cdot|s_t)$. Note that we chose to use the same triplets for all policies since we use off-policy algorithms, which allows us to use the acquired data more efficiently.

## 6 EXPERIMENTS

We performed a series of challenges to see if DESTA **1)** learns to perform *safe planning* **2)** learns to select the appropriate state to perform safe override interventions which avoids a trade-off between safety and task performance **3)** learns to use deterministic controls to ensure precise actions. In these tasks, we compared the performance of DESTA with state-of-the-art RL methods for safe learning: PPO (Schulman et al., 2017), Lagrangian PPO, TRPO (Schulman et al., 2015), CPO (Achiam et al., 2017), and WCPG (Tang et al., 2019). We then compared DESTA against these baselines on performance benchmarks in challenging high dimensional problems in safety gym benchmarks (Yang et al., 2021) which simulate real-world physical environments and OpenAi's Lunar Lander.

**Experiment 1. Safe Planning**: Having an individual agent that is responsible for safety enables our method learn to *plan ahead* to minimise safety violations. In particular, it is able to learn to take actions to avoid safety violations at future states. To demonstrate this, we designed a T-Junction environment with two goal states. The environment is a T-shaped grid of three corridors. The agent starts at one end of the neutral corridor and must advance to the junction at the other end. From here it must choose to go left or right into a safe or an unsafe corridor respectively. They each have a reward at the far end but the unsafe corridor additionally has a safety cost. In our experiments, the safe goal has a reward of 50, the unsafe goal a reward of 100 and all other squares a one-shot reward of 10. Each square in the unsafe corridor has a 10% chance of giving a safety cost of 100. Therefore, the agent must learn to forego a larger reward to stay safe, with Safety Agent directing it down the safe corridor. The maximum possible safe return per episode is 170, any higher and the unsafe corridor was breached.

We compared our algorithm to PPO, Lagrangian PPO, TRPO, Lagrangian TRPO, SAC and CPO. DESTA-SAC (ours), Lagrangian PPO and Lagrangian TRPO were able to solve the environment by going down the correct corridor, with the others turning into the unsafe one. Ours trains faster and is more stable across seeds than the other baselines that solve the environment.

**Experiment 2. Safe precision control using the Lunar Lander (Brockman et al., 2016)**: Since Safety Agent uses deterministic controls to perform its actions, Safety Agent is able to perform precise actions to ensure the safety of the system. This allows our method to avoid safety violations in instances where precision is required. To verify this claim, we used the Lunar Lander environment within OpenAI gym (Brockman et al., 2016). Given that there is no strict definition of safety violation as in the safety-gym environments (Brockman et al., 2016), we manually defined that a safety violation is incurred whenever the spacecraft transitions outside a fixed horizontal threshold radius from the origin. By introducing such safety violation our goal was to test if DESTA can exploit the interplay between the reward maximisation and cost minimisation (i.e., ensure the spacecraft is always on the near-optimal path), hence enabling stronger and more efficient learning. We include additional experiment details in the Appendix.

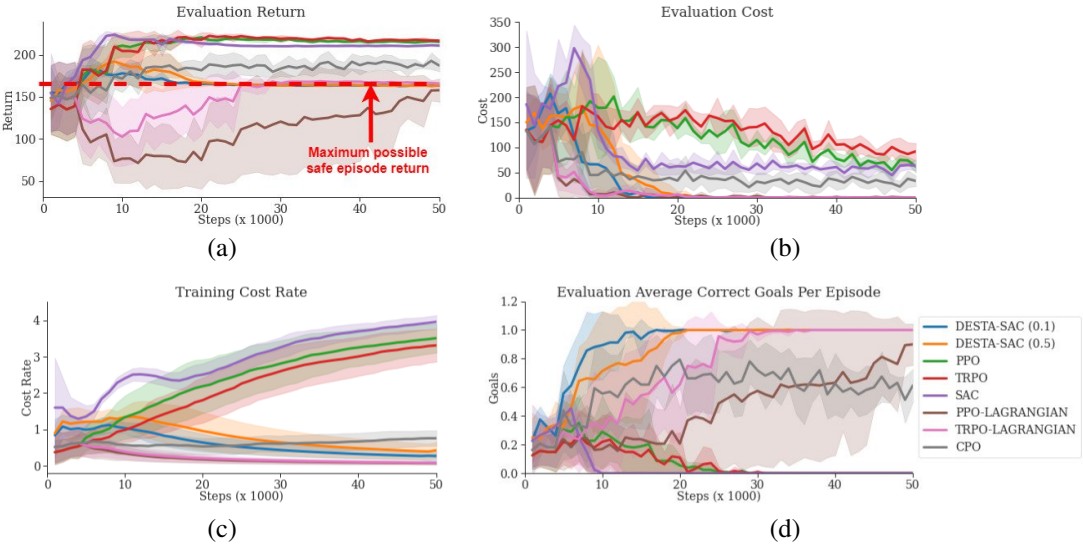

(a)                                                           (b)

(c)                                                           (d)

Figure 2: Empirical evaluation of DESTA and baselines on T-Junction environment. 32 evaluation episodes were run every 1000 steps and 5 seeds were used per algorithm. Intervention costs of both 0.1 and 0.5 were used and learning rates were 0.001. Note that 20 SAC gradient steps were required to achieve the reported performance. (a) Average return of evaluation episodes. On this environment, 170 is the maximum safe return per episode, obtained by moving through all centre squares, the junction square and all squares in the safe corridor without turning into the unsafe corridor. Any higher means the unsafe corridor was entered. (b) Average cost of evaluation episodes. (c) Average training cost per step so far. (d) Average number of correct (left) goals per evaluation episode.

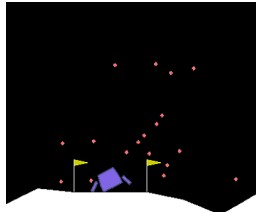

Figure 3: The lander must land on the pad between two flags.

In Fig. 4(a) we observe that the DESTA agent outperforms all the baseline agents, in terms of both the sample efficiency and the asymptotic score. Moreover, we observe that DESTA enables more stable training, possibly due to the reduced action space given the intervention of the Safety Agent. In Fig. 4(b) we show the accumulative safety violation cost throughout the course of training, and we again observe that DESTA yields the lowest accumulative cost amongst all the evaluated agents, whilst maintaining the stability of the cost across different random seeds. Such behaviour indicates that the Safety Agent is well-trained such that the safety-violation states are much more rarely encountered comparing to the baseline agents.

We note that the consistent low variance across various independent training runs indicates that DESTA enables low sensitivity with respect to the randomness given different random seeds that plagues many of the existing RL algorithms (Colas et al., 2018). We also examined the behaviours of the intervention through the course of training (Fig. 8), which nicely corresponds with our intuition that the interventions mostly occur during the early phase of each episode. See the Appendix 10 for further discussion.

**Safety Gym Benchmark Experiments:** We compared DESTA with the baselines in challenging environments within a Safety Gym Benchmark in (Yang et al., 2021). These environments, each representing simulated physical systems vary in the number and size of hazards, and generation of goal and hazards' locations. In the first experiment, the hazards are static. We then tested in a setting with moving hazards. In terms of performance, DESTA outperforms the best competing baselines.

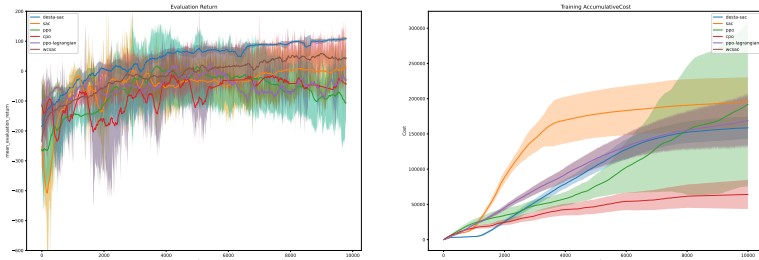

Figure 4: Empirical evaluation of DESTA on the LunarLander task. (a) Evaluation of the agents on the LunarLander task. (b) Accumulative safety violation cost during training.

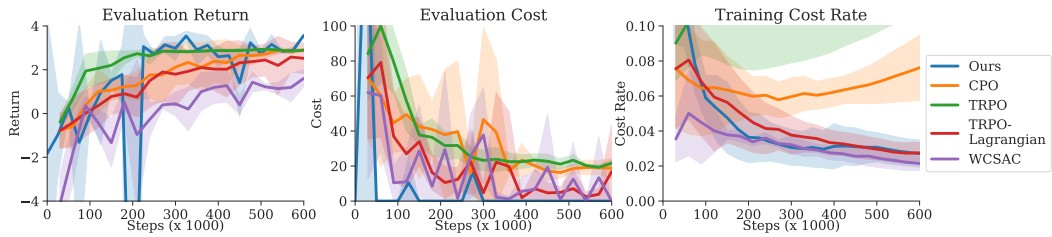

Figure 6: StaticEnv-v0

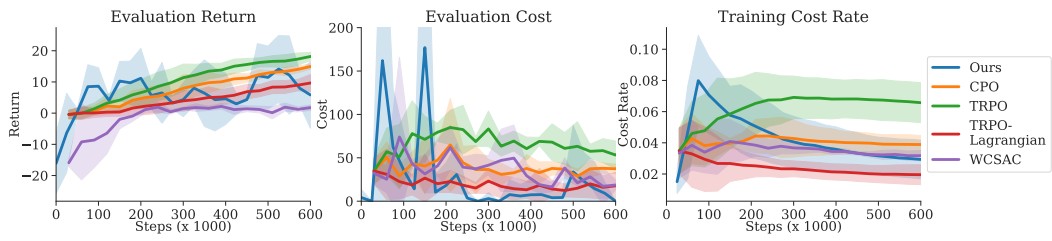

Figure 7: DynamicEnv-v0

In terms of safety violations, DESTA incurs a lower number of violations as compared to the best competing baselines.

# 7 CONCLUSION

In this paper, we presented a novel game theoretical framework to solve the problem of learning safely. Our Markov game framework of a Task agent and a second Safety agent decouples the tasks of ensuring safety and maximising the task reward. This enables one of the agents to learn complex behaviours at subsets of states to ensure the goal of minimising safety violations and at the same time exhibiting adaptive behaviour to the

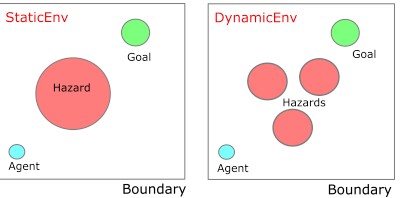

Figure 5: Safety Gym Benchmark environments. Left: Static environment with one hazard. Right: Dynamic environment with multiple hazards.

Task agent that maximises the environment reward at all other states. By presenting a theoretically solid and empirically robust approach to solving the safe learning problem, our method opens up the applicability of RL to a range of real-world control problems with sophisticated safety constraints. In future, we believe our approach of marrying RL, MARL and game can be adopted to solve other open challenges in RL.

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

# Part I

# Appendix

# Table of Contents

# 8 HYPERPARAMETER SETTINGS

Table 1: Hyperparameters for DESTA.

| | | T junction | Lunar lander | StaticEnv | DynamicEnv |
|---|---|---|---|---|---|
| **Runner** | # gradient steps | 0 | 0 | 8 | 8 |
| | # environment steps | 0 | 0 | $600k$ | $600k$ |
| | Agent frequency update | 0 | 0 | 4 | 4 |
| | Agent batch size | 0 | 0 | 64 | 64 |
| **Agents** | Policy network dimensions | $\{0\}$ | $\{0\}$ | $\{256, 256\}$ | $\{256, 256\}$ |
| | Policy networks activations | ReLU | ReLU | ReLU | ReLU |
| | Value network layer dims | $\{0\}$ | $\{0\}$ | $\{256, 256\}$ | $\{256, 256\}$ |
| | Value networks activations | ReLU | ReLU | ReLU | ReLU |
| | Discount factor | 0 | 0 | 0.99 | 0.99 |
| | Polyak update scale | 0 | 0 | 0.005 | 0.005 |
| | Intervention cost | 0 | 0 | 0.1 | 0.5 |
| **Optimiser** | Opt Algorithm | Adam | Adam | Adam | Adam |
| | Policy learning rate | 0 | 0 | $10^{-4}$ | $10^{-4}$ |
| | Value function learning rate | 0 | 0 | $10^{-4}$ | $10^{-4}$ |
| | Temperature learning rate | 0 | 0 | $10^{-4}$ | $10^{-4}$ |

## 9   PROOF OF TECHNICAL RESULTS

### PROOF OF PROPOSITION 1

**Proof:**   [Proof of Prop. 1] The proof is given by establishing a contradiction. Therefore suppose that $\mathcal{M}^{\pi,\pi^2}\psi(s_{\tau_k}, I(\tau_k)) \leq \psi(s_{\tau_k}, I(\tau_k))$ and suppose that the intervention time $\tau_1' > \tau_1$ is an optimal intervention time. Construct the safety agent $\pi'^2 \in \Pi^2$ and $\tilde{\pi}^2$ policy switching times by $(\tau_0', \tau_1', \dots,)$ and $\pi'^2 \in \Pi^2$ policy by $(\tau_0', \tau_1, \dots)$ respectively. Define by $l = \inf\{t > 0; \mathcal{M}^{\pi,\pi^2}\psi(s_t, I_0) = \psi(s_t, I_0)\}$ and $m = \sup\{t; t < \tau_1'\}$. By construction we have that

$$v_2^{\pi,\pi'^2}(s, I_0)$$
$$= \mathbb{E}\left[R(s_0, a_0) + \mathbb{E}\left[\dots + \gamma^{l-1}\mathbb{E}\left[R(s_{\tau_1-1}, a_{\tau_1-1}) + \dots + \gamma^{m-l-1}\mathbb{E}\left[R(s_{\tau_1'-1}, a_{\tau_1'-1}) + \gamma\mathcal{M}^{\pi,\pi'^2}v_2^{\pi,\pi'^2}(s', I(\tau_1'))\right]\right]\right]\right]$$
$$< \mathbb{E}\left[R(s_0, a_0) + \mathbb{E}\left[\dots + \gamma^{l-1}\mathbb{E}\left[R(s_{\tau_1-1}, a_{\tau_1-1}) + \gamma\mathcal{M}^{\pi,\tilde{\pi}^2}v_2^{\pi,\pi'^2}(s_{\tau_1}, I(\tau_1))\right]\right]\right]$$

We now use the following observation $\mathbb{E}\left[R(s_{\tau_1-1}, a_{\tau_1-1}) + \gamma\mathcal{M}^{\pi,\tilde{\pi}^2}v_2^{\pi,\pi'^2}(s_{\tau_1}, I(\tau_1))\right]$

$$\leq \max\left\{\mathcal{M}^{\pi,\tilde{\pi}^2}v_2^{\pi,\pi'^2}(s_{\tau_1}, I(\tau_1)), \max_{a_{\tau_1}\in\mathcal{A}}\left[R(s_{\tau_k}, a_{\tau_k}) + \gamma\sum_{s'\in\mathcal{S}}P(s'; a_{\tau_1}, s_{\tau_1})v_2^{\pi,\pi^2}(s', I(\tau_1))\right]\right\}.$$

Using this we deduce that

$$v_2^{\pi,\pi'^2}(s, I_0) \leq \mathbb{E}\left[R(s_0, a_0) + \mathbb{E}\left[\dots\right.\right.$$

$$+ \gamma^{l-1}\mathbb{E}\left[R(s_{\tau_1-1}, a_{\tau_1-1}) + \gamma\max\left\{\mathcal{M}^{\pi,\tilde{\pi}^2}v_2^{\pi,\pi'^2}(s_{\tau_1}, I(\tau_1)), \max_{a_{\tau_1}\in\mathcal{A}}\left[R(s_{\tau_k}, a_{\tau_k}) + \gamma\sum_{s'\in\mathcal{S}}P(s'; a_{\tau_1}, s_{\tau_1})v_2^{\pi,\pi^2}(s', I(\tau_1))\right.\right.\right.$$

$$= \mathbb{E}\left[R(s_0, a_0) + \mathbb{E}\left[\dots + \gamma^{l-1}\mathbb{E}\left[R(s_{\tau_1-1}, a_{\tau_1-1}) + \gamma\left[Tv_2^{\pi,\tilde{\pi}^2}\right](s_{\tau_1}, I(\tau_1))\right]\right]\right] = v_2^{\pi,\tilde{\pi}^2}(s, I_0))$$

where the first inequality is true by assumption on $\mathcal{M}$. This is a contradiction since $\pi'^2$ is an optimal policy for safety agent. Using analogous reasoning, we deduce the same result for $\tau_k' < \tau_k$ after which deduce the result. Moreover, by invoking the same reasoning, we can conclude that it must be the case that $(\tau_0, \tau_1, \dots, \tau_{k-1}, \tau_k, \tau_{k+1}, \dots,)$ are the optimal intervention times.   $\square$

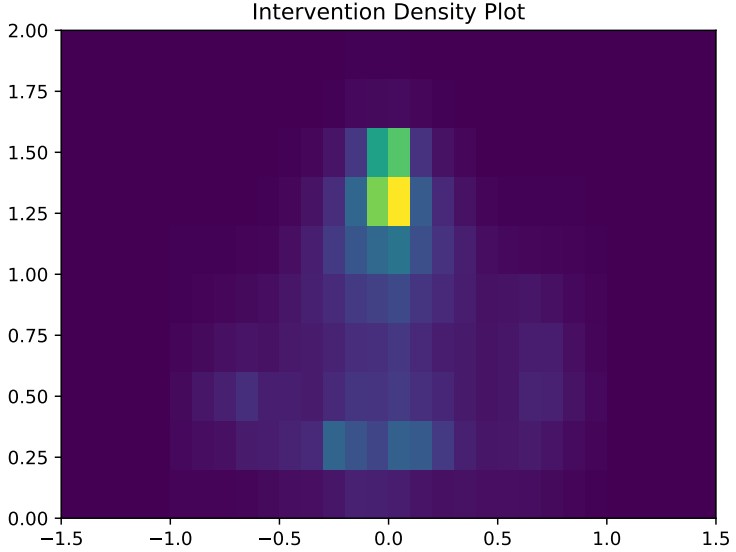

Figure 8: Histogram of intervention location during training.

## 10   LUNARLANDER EXPERIMENT HISTOGRAM

