# OpenReview forum: "DESTA: A Framework for Safe Reinforcement Learning with Markov Games of Intervention"
_ICLR.cc/2022/Conference — ICLR 2022 Submitted_

### Official Review · Reviewer_mFYr · 2021-10-28

**Correctness:** 2
**Technical Novelty And Significance:** 2
**Empirical Novelty And Significance:** 2
**Recommendation:** 3
**Confidence:** 2

**Details Of Ethics Concerns:**

I don't have ethical concern.

**Main Review:**

This paper is easy to read, and the key ideas are clearly presented.

As far as the reviewer knows, it is a new idea to formulate safe RL problems into Markov games with two players (i.e., Task Agent and Safety Agent). This framework enables us to train an agent more easily compared with Lagrangian methods since reward and safety signals are decoupled.

I have several concerns below.

First concern lies in Experiment. The authors conducted experiments in three different environments, but I do not think that the effectiveness and merits of the proposed method (framework) are not shown. For example, in Figure 2, how do the authors convince the readers that their proposed method over-performs the other baselines. If the authors are concerned about many constraint violations in the early training phase, then DESTA's results presented in Figure 7 is also problematic.
Given there is no theoretical result regarding optimality or safety guarantees, I think that merits of the proposed method are neither presented theoretically nor empirically.

Second concern is advantages over Lagrangian methods (e.g., TRPO-Lagrangian), which is also related to the first concern.
In the Section 1, the authors criticizes Lagrangian-based method saying "Consequently, the resulting policies do not offer guarantees of safety during training and can
lead to safety violations both during and after training". However, seeing the experimental results, the proposed method also violates safety constraints in a similar level compared with Lagrangian-based RL algorithm. I am not fully convinced whether the proposed method is actually better than existing methods.

The last concern is reproducibility. Since there is no source-code attached and only little information is provided in the appendix, I do not think the experimental results presented in this paper is sufficiently reproducible. For example, how did you implement baselines and what is the parameters?

Minor comments
- Baselines are different for each experiment (for example, WCSAC is not tested in T-junction environment and TRPO-Lagrangian are not tested in LunarLander). Are there any reasons why such baselines are omitted?
- In section 2, the action space for agent $i$ is defined as $\mathcal{A}_i$, but in Section 3A, the it is denoted as $\mathcal{A}^i$. Also, the action space for Task Agent is somewhat abbreviated into $\mathcal{A}$ ($i$ is omitted). Please make such definitions consistent throughout the paper.
- In the main text (Section 5), the authors say that WCPG is a baseline, but in Figures 4 and 5, results for WCSAC is presented.
- Please add shade (e.g., standard deviation) in Figure 2.

**Summary Of The Paper:**

This paper proposes a new framework for safe RL which is called DESTA. DESTA is a framework where two players called Safety Agent and Task Agent interact with the environment. The task of Safety Agent is to minimize the violations of the safety constraints and tat of Task Agent is to maximize the cumulative reward. Safety of the (safety-agnostic) Task Agent is encouraged by Safety Agent's interventions. The authors compare the performance of their proposed method with state-of-the-art RL baselines in three tasks.

**Summary Of The Review:**

Though DESTA framework is new and interesting, its merits are neither presented theoretically nor empirically. In addition, although there is no theoretical results supporting the authors' claims, reproducibility is rather low.
Hence, I vote for rejection.

---

### Official Review · Reviewer_vp9k · 2021-11-02

**Correctness:** 2
**Technical Novelty And Significance:** 3
**Empirical Novelty And Significance:** 2
**Recommendation:** 3
**Confidence:** 3

**Main Review:**

### Strengths:

The idea of using Markov games to model the problem of safe-exploration is particularly interesting and novel to the best of my knowledge.

### Concerns:

- *Missing related-work:* The idea of having decoupled task objectives and safety planning has been explored in the community, although not from the lens of Markov games. Some of the works that come to mind are: [1] where barrier functions take the role of safety planner (Safety Agent) and can override the Task Agent policy,  [2] where pessimism is used for the Safety planning, and even in CPO based approached where there is a fallback policy when the agent violates the safety constraint that overrides the original task policy [3].  I would expect some more comparisons or details on the trade-offs of the proposed approach with the related work.

- *Intervention cost origins:* In the safety objective each intervention occurs a  cost $c$ (or $\kappa$) that is not part of the original CMDP formulation. How do we get this cost function? Is it a hyper-parameter or design-time decision? If so, we will have the same limitations as the Lagrangian methods. What’s the benefit of the proposed approach then?
The authors mention that unlike other work based on interventions they don’t need to design manually engineered safety responses, but I will argue that designing a cost function for intervention is just another form for that.

- *Confusing writing:* I found the writing for Section 3 particularly confounding. The authors add on notations and terms in a very casual manner. I would expect formal definitions for each of the introduced new terms.

- *Implication of Proposition 1?:* Proposition just tells us there exists a solution policy corresponding to the intervention reward. If we consider that the intervention reward function is a linear combination of different reward functions (with the same underlying transition) then isn’t this expected? I guess I’m failing to understand the importance of Proposition 1.

- *Baselines:* Note that there are approaches like [4] that are not based on Lagrangian formulation and still can be used in the context of CMDPs during the learning, with even better performance than Lagrangian formulations. They would have made stronger baselines and would have strengthened the claim.

- *Environments:* I would like to know why the authors decided to not use any of the environments that are used in context with the baselines (for instance OpenAI Safety Gym for Lagrangian-PPO, CPO etc.). Only the simpler environment (T-maze) is stochastic where both the lagrangian and proposed approaches perform similarly. The other environments considered in this work seem deterministic and not sufficient to support the claims.

- *Reproducibility:* As all the claims are empirical, providing the accompanying code would have helped to strengthen them. There is also no mention of the number of seeds. Fig 6,7 have no captions or info about the axis.



**Summary Of The Paper:**

 The paper presents an alternate approach to the problem of minimizing constraint violations during learning in CMDPs (safe exploration). They formulate the problem using the framework of Markov games, where three policies are learned: (i) a policy that maximized just the primary reward objective for CMDP, (ii) a safety or override policy that is deterministic and focuses on minimizing the constraint violation, and (iii) an intervention policy that switches between (i) and (ii). The authors then test their approach on toy environments: T-maze, Lunar-lander and Safety Gym Benchmark.

**Summary Of The Review:**

Although the paper has some interesting ideas, it needs significantly more work in strengthening the supporting claims (weak experiments as well as no code), placing the work in context to the related work, as well as writing and presentation. I think the paper in its current state will be marginally useful to the community, and as such vote for rejection.

---

### Official Review · Reviewer_BFBF · 2021-11-04

**Correctness:** 3
**Technical Novelty And Significance:** 2
**Empirical Novelty And Significance:** Not applicable
**Recommendation:** 5
**Confidence:** 2

**Main Review:**

Strengths:

(1). The paper proposes a Markov game-based method for dealing with two competing objects: reward maximization and constraint satisfaction. This method incorporates the idea of the Markov game into the constrained Markov decision processes. In my view, it is a new and interesting idea.

(2). The proposed method is flexible in many aspects, e.g., dealing with reward/cost separately, using a deterministic policy, and incorporating different safety criteria.

(3). The paper also provides comparison experiments to demonstrate the performance of that the proposed method.


Weaknesses:

(1). It is important to specify the meaning: reward maximization and the safety violation minimization are fully decoupled. Can you provide any evidence on this in the proposed framework?

(2). I believe the framework in Section 3 is a key contribution. Is it always true that we can transform an expectation constraint into the objective for the safety agent? How exactly is it?

(3). Do you have any theory that supports the proposed method? I feel the authors may devote some efforts to analyzing the proposed general-sum Markov games.

**Summary Of The Paper:**

The authors use a Markov game-based approach to study constrained Markov decision processes, propose a safety training algorithm, and show better performance than baseline methods in computational experiments.


**Summary Of The Review:**

Overall, I like the idea in the paper. However, I won't be able to support the acceptance. My major concern is theoretical support and clarity. Hopefully, the authors can supplement some progress during the rebuttal.

---

### Official Review · Reviewer_7tNg · 2021-11-06

**Correctness:** 2
**Technical Novelty And Significance:** 3
**Empirical Novelty And Significance:** 3
**Recommendation:** 3
**Confidence:** 2

**Main Review:**

Safe RL problems are very important in many applications. The paper considers the c-MDP formulation for safe RL and proposes a novel Markov game approach, called DESTA, to solve c-MDP. The idea of DESTA is to decompose a c-MDP agent into two agents: the task agent and the safety agent. The action of the c-MDP agent is selected by either the task agent or the safety agent depending on who is currently in control, and the safety agent also determines which agent is in control at each time. With these two agents, the c-MDP is transformed into a Markov game between the two agents. Given the safety agent's policy, the task agent try to maximize rewards. Given the task agent's policy, the safety agent try to determine when to take over control and what actions to take in order to minimize costs.

This idea of formulating a Markov game between two agents to tackle c-MDP is very interesting. However, despite making some intuitive sense, the paper has no analysis on why the "solution" of the Markov game may be good for the c-MDP. There are two major issues with this Markov game framework.

First, the paper never discusses what is the "solution" of the game or what is the goal the algorithm is trying to achieve after learning from the game. Since the two agents have different objective, typically one may seek to find an equilibrium of the game. But since the game is none zero-sum, there could be multiple equilibria in general. It is even more complicated when the policy of the safety agent is further decomposed in to an action selection policy and an intervention policy. So in fact there are three agents with three different objective functions. Does the policies generated by the algorithm converge to some kind of equilibrium of the game? Does the proposed training method has some convergence properties?

Second, even if the algorithm converges to some equilibrium policies, it's not clear if there is any connection between these policies and the c-MDP performance. Does the two separate policies for the safety agent trained by different objectives minimize the cost for the safety agent? Does the resulting overall policy satisfies the safety constraints? Are there reward guarantees using the equilibrium task and safety policies?

Another issue of the paper is the confusing notations. In many parts of the paper the interventions are represented by a sequence of intervention times. These intervention times creates many unnecessary lengthy equations with summations over mostly zero terms. In fact, the intervention could be represented by whether $a^{int}_t$ is zero or one, as described in the algorithm pseudo code. The current notations create several readability issues, and some of the equations do not make sense. For example, $Q_2^{\pi, \pi^{2, safe}}$ has two different definitions, one in Definition 1 and one in Proposition 1.

For the experiments, although the paper claims that the proposed method performs the best, it actually performs worse than some baselines in Figure 7. The abrupt changes in the return in Figure 6 and in the cost in Figure 7 also indicates some stability concerns.

**Summary Of The Paper:**

The paper proposes Markov game based framework for safe reinforcement learning. The proposed method consists of two agents: the task agent selecting actions to maximize the reward, while the other safety agent determining whether to take over control of actions to minimize the cost. Experimental results are shown in three simulated environments.

**Summary Of The Review:**

Pros
*  Novel and interesting idea to use the two agents Markov game to tackle c-MDP.

Cons
* Some issues with the Markov game framework and no analysis for them.
* Confusing notations and problematic formulas.
* Experimental results are mixed.

---

### Decision · Program_Chairs · 2022-01-20

**Decision:**

Reject

**Comment:**

This paper investigated online safe reinforcement learning problem in the constraint MDP setting. By introducing Safety Agent and Task Agent, the authors translate the RL problem into a Markov game. The AC agrees with all reviewers that there is a lack of theoretical analysis and experimental comparisons with existing benchmarks. It has not reached the bar of ICLR papers.